# Moringa Oleifera Alleviates Aβ Burden and Improves Synaptic Plasticity and Cognitive Impairments in APP/PS1 Mice

**DOI:** 10.3390/nu14204284

**Published:** 2022-10-14

**Authors:** Yacoubou Abdoul Razak Mahaman, Jun Feng, Fang Huang, Maibouge Tanko Mahamane Salissou, Jianzhi Wang, Rong Liu, Bin Zhang, Honglian Li, Feiqi Zhu, Xiaochuan Wang

**Affiliations:** 1Co-Innovation Center of Neuroregeneration, Nantong University, Nantong 226001, China; 2Cognitive Impairment Ward of Neurology Department, The Third Affiliated Hospital of Shenzhen University, 47 Youyi Rd., Shenzhen 518001, China; 3Department of Pathophysiology, School of Basic Medicine, Key Laboratory of Education Ministry and Huibei Province of China for Neurological Disorders, Tongji Medical College, Huazhong University of Science and Technology, Wuhan 430030, China; 4Department of Neurosurgery, Union Hospital, Tongji Medical College, Huazhong University of Science and Technology, Wuhan 430022, China; 5College of Health, Natural and Agriculture Sciences Africa University, Mutare P.O. Box 1320, Zimbabwe; 6Department of Pathology and Pathophysiology, School of Medicine, Jianghan University, Wuhan 430056, China; 7Shenzhen Research Institute, Huazhong University of Science and Technology, Shenzhen 518000, China

**Keywords:** Alzheimer’s disease, amyloid beta, synaptic plasticity, cognitive impairment, *Moringa oleifera*

## Abstract

Alzheimer’s disease is a global public health problem and the most common form of dementia. Due to the failure of many single therapies targeting the two hallmarks, Aβ and Tau, and the multifactorial etiology of AD, there is now more and more interest in nutraceutical agents with multiple effects such as *Moringa oleifera* (MO) that have strong anti-oxidative, anti-inflammatory, anticholinesterase, and neuroprotective virtues. In this study, we treated APP/PS1 mice with a methanolic extract of MO for four months and evaluated its effect on AD-related pathology in these mice using a multitude of behavioral, biochemical, and histochemical tests. Our data revealed that MO improved behavioral deficits such as anxiety-like behavior and hyperactivity and cognitive, learning, and memory impairments. MO treatment abrogated the Aβ burden to wild-type control mice levels via decreasing BACE1 and AEP and upregulating IDE, NEP, and LRP1 protein levels. Moreover, MO improved synaptic plasticity by improving the decreased GluN2B phosphorylation, the synapse-related proteins PSD95 and synapsin1 levels, the quantity and quality of dendritic spines, and neurodegeneration in the treated mice. MO is a nutraceutical agent with promising therapeutic potential that can be used in the management of AD and other neurodegenerative diseases.

## 1. Introduction

The increase in longevity as a result of scientific progress leads to the increasing aging of the population globally. This results in an increase in the prevalence of chronic diseases such as diabetes and neurodegenerative diseases such as Alzheimer’s disease (AD). AD is the most prevalent neurodegenerative disease and the most common cause of dementia, increasing the health burden worldwide and impairing the memory, language, and thinking abilities of the affected individual [1]. Unless therapeutic strategies are found, the burden of this major public health problem in modern society will exponentially increase in the coming years. AD is histopathologically marked by the extracellular senile plaques and intracellular neurofibrillary tangles (NFTs) which are predominantly made up of amyloid-β (Aβ) peptides and hyperphosphorylated Tau, respectively. The Aβ peptides are the result of proteolysis of the amyloid precursor protein (APP) by two proteases, the beta APP cleaving enzyme 1 (BACE1), also called β-secretase, and the γ-secretase [2,3,4], while the hyperphosphorylated Tau protein in the NFTs results from the dysregulation of kinase/phosphatase system [5]. A substantial amount of evidence from previous studies has established that Aβ is the leading cause or trigger in the AD pathogenesis [6,7,8,9,10,11]. BACE1 is the rate-limiting enzyme in the amyloidogenic APP processing and therefore plays an important role in Aβ generation [12]. Moreover, it is believed that in sporadic AD, an age-associated decrease in Aβ clearance is also critical for Aβ accumulation [13,14,15,16]. Therefore, inhibiting Aβ generation and enhancing its clearance might synergically contribute to the improvement of AD. However, many of the Aβ-targeted therapies have failed [17,18,19,20]. This failure and the multifactorial complexity of AD trigger therapeutic interest in products with multiple effects such as nutraceuticals such as *Moringa oleifera* (MO), *Tamarix gallica, Ginkgo biloba*, *Codonopsis pilosula* polysaccharide, and Ferulic acid, which have shown potential antioxidative, anti-inflammatory, and neuroprotective virtues among others [21,22,23,24,25,26,27].

In the clinic, the age-dependent learning and memory impairments that culminate into cognitive deficits, and the accompanying synapse loss are the most common characteristics of AD patients [28] and many AD animal models [29]. Synaptic dysfunction also correlates with the degree of cognitive decline in AD patients and transgenic animals with Aβ toxicity [28]. NMDA receptors (NMDARs) are key players for synaptic plasticity and thus learning and memory [30] and Aβ is reported to decrease GluN2B positive synaptic spines as well as the surface expression of GluN2B containing NMDA receptors [31]. Moreover, animal models of AD also showed reduced surface expression and dephosphorylation of the GluN2B subunit of NMDAR at Tyr1472, which correlated with the receptor endocytosis [31,32]. Interestingly, striatal-enriched protein tyrosine phosphatase (STEP), the main phosphatase of GluN2B and GluA2 subunits of NMDA and AMPA receptors, as well as of synapse-related kinases, Fyn, Pyk2, and ERK1/2, was reported to be increased in AD including in post-mortem AD patients and several AD mice models such as the Tg2576, J20, APP/PS1, and 3×TG mice [32,33,34]. Interestingly, inhibition of STEP, either pharmacologic [35] or genetic [33], was reported to ameliorate cognitive function and hippocampal memory in the 3×Tg-AD mouse model, as well as restored GluN2B phosphorylation at Tyr1472 [35]. Fyn is the main kinase that phosphorylates GluN2B at Tyr1472 [36,37,38] and Fyn activity is regulated by its phosphorylation at Tyr416. Interestingly, STEP can dephosphorylate (inactivate) Fyn at Tyr416 [39], thus impairing Tyr1472 GluN2B phosphorylation. On the other hand, in AD Aβ can induce an increase in STEP via impairing the ubiquitin-proteosome system as well as via activating the α7 nicotinic acetylcholine receptors (𝛼7nAChRs) [40,41,42] which leads to intracellular calcium influx, activating calcineurin (protein phosphatase 2B (PP2B)). The active PP2B can subsequently dephosphorylate (inactivate) the inhibitor of protein phosphatase 1 (PP1), DARPP-32 at Thr34, thereby leading to the dephosphorylation (activation) of PP1 at Thr320. Interestingly, PP1 can undergo auto-dephosphorylation and can trans-dephosphorylate other PP1 molecules [43,44]. Thus, upon removal of the inhibitory effect of DARPP-32, the active PP1 can then dephosphorylate (activate) STEP at Ser221 [32].

*Moringa oleifera* is one of the natural compounds that contain many active constituents including flavonoids, alkaloids, glycosides, lipids, proteins, carbohydrates, minerals, and vitamins among others [45,46] and belongs to the family of *Moringaceae* that are distributed widely in many African and Asian countries [47]. It has been reported to have antioxidative, anti-inflammatory, antimicrobial, bactericidal, hypoglycemic, anti-cancer, anti-aging, and neuroprotective potentials [48,49,50,51,52,53,54,55]. In the field of AD, MO was reported to exert anti-aging, antioxidative, anti-cholinesterase, and neuroprotective activities [27,49,56,57,58] and is safe at higher doses in both rats and mice as its LD50 was more than 6400 mg/kg in mice [59,60,61]. Moreover, in our previous study [21], we found that MO can alleviate homocysteine-induced AD-like pathological changes including Aβ production, Tau hyperphosphorylation, and neurodegeneration via decreasing BACE1, GSK-3β, CDK5, CaMKII, increasing PP2A activities, and improving oxidative stress and cognitive deficits. However, whether MO has similar effects on APP/PS1, the Aβ model of AD remains enigmatic. Therefore, in this study, we evaluate the effect of MO extract on Aβ load, synaptic plasticity, and learning and memory in APP/PS1.

## 2. Materials and Methods

### 2.1. Animals

Male APP/PS1 mice 3 months of age were purchased from Cavins Laboratory Animal Co., Ltd. (Changzhou, China), and the C57BL/6J wild-type (WT) mice were purchased from the Experimental Animal Centre of Tongji Medical College. Four to five animals were housed in ventilated cages, in a thermoregulated and pathogen-free environment. The mice were maintained under a 12/12-h light–dark cycle and had free access to food and water. All experiment protocols for the mice were according to the Huazhong University of Animal Care and Use Committee (protocol code (2020) IACUC Number:2735).

### 2.2. Moringa oleifera (MO) Extraction and Treatment

MO leaf powder was purchased from Moringa Smart (China). The powder was exhaustively macerated in 80% methanol as previously described [21]. Briefly, 100 g of the MO powder was soaked in 500 mL of 80% methanol and allowed at 4 °C with continuous shaking for 2 days. The extract was then filtered through Whatman filter paper and concentrated using a vacuum rotary evaporator at 40 °C. The condensed final yield of the extract was dark green and was kept in a deep freezer at −80 °C until use.

Three-month-old male APP/PS1 and WT mice were divided into 3 groups: the WT as control, APP/PS1 as the AD model, and APP/PS1 with MO treatment as the treatment group. During the one-week acclimatization period of the mice, their water intake per day was evaluated and was estimated to be 4–5 mL. The MO extract was then resuspended in drinking water to a final concentration of 400 mg/kg/day in 5 mL so that the mice have an average consumption of 400 mg/kg of MO every day. The AD model (APP/PS1) and the WT (C57BL/6J) mice groups were provided normal drinking water without MO. The bottles of water were changed every day and the treatment lasted for four months.

### 2.3. Behavioral Tests

The animals were randomly subjected to each behavioral test. Between each animal and trial, the apparatuses for every behavioral test were wiped every time with 75% ethanol. The scoring was performed by researchers that are blinded to both the genotypes and treatments of the animals and the testing was performed in a dimly lit room.

#### 2.3.1. Morris Water Maze (MWM) Test

The MWM test was employed to evaluate the spatial learning of mice and was carried out as previously described [62]. Briefly, the mice were habituated to the behavior room before the test began, and the acquisition test was carried out for 6 days, 3 trials per day with each trial lasting 60 s. When a mouse finds and climbs onto the platform within 60 s, it would then be allowed to stay on the platform for 20 s; otherwise, at the end of the 60 s, it would be gently guided to the platform and allowed to stay for 20 s. The latency (s) to find the hidden platform was recorded after each trial of every learning session. Then twenty-four hours after the training ended, the hidden platform was removed, and the probe trial was performed for 60 s, then the latency to cross the platform location and the number of platform crossings, as well as the time in the target quadrant and the total distance covered were all recorded.

#### 2.3.2. Open Field Test (OFT)

To evaluate anxiety-like behaviors and spontaneous movements of the mice we used the OFT. The apparatus (Techman Software Co., Ltd., Chengdu, China) used for this test was fabricated of a plastic container constituting an open field arena (50 × 50 × 50 cm), while a digital camera was hung directly above the center of the field. The square field was divided into 5 × 5 zones and the middle 3 × 3 sectors were defined as the center area. The mice were gently placed one by one in the open field arena and allowed to freely explore for 5 min. The center duration and the total distance traveled were both recorded by the camera system.

#### 2.3.3. Novel Objective Recognition Test (NOR)

This test is used to evaluate the ability of the mice to recognize an old object and show a preference for exploring a new object. For this, the mice were placed for 5 min in the apparatus, made up of a 50 × 50 × 50 cm plastic container, with no object and were allowed for 24 h in the test room before the experiment was carried out. On the first testing day, objects A and B, different in shapes and colors, were placed in two of the corners of the box and the mice were brought into the middle of the area and were allowed to explore objects A and B for 5 min. After 24 h i.e., on the second day, object B was replaced by object C of a different shape and color from A and B. The exploration time of the mice on objects A, B, and C was recorded, and the recognition index was calculated as TA/(TA + TC) and TC/(TA + TC). TA and TC represent the exploring time of the mice on objects A (old) and C (new), respectively.

#### 2.3.4. Fear Conditioning Tests

This helps to assess the contextual and cue memory and the experiment was performed as previously described [62]. Briefly, the mice were first habituated for 3 min in the test chamber and on the first day, a conditioned stimulus (CS) was delivered and was immediately followed by an unconditioned stimulus (US). Then on the second day, i.e., 24 h after conditioning, the context-dependent test was performed by taking the mice back to the same training chamber for 5 min with no CS or US. The cue-dependent test was carried out on the third day, i.e., 48 h after conditioning, with different contextual cues, and each mouse was allowed for 5 min in total without the US, but after 2 min of free exploration, the mouse was exposed to the exact same CS. The freezing responses in both the context and cue conditions were recorded.

### 2.4. Golgi Staining

After anesthesia and perfusion with normal saline, the brains of the mice were collected and directly placed in Golgi solution (1 g potassium dichromate, 1 g mercuric chloride, 0.8 g potassium chromate, and 100 mL double-distilled water) for 2 weeks in the dark with the solution being changed every 2 days. The brains were then sequentially incubated in 10%, 20%, and 30% sucrose protected from light and sectioned at 100 μm thickness with a vibratome (Leica, VT1000S, Nussloch, Germany) and then placed on gelatin-coated glass slides. After rinsing with double-distilled water, the slides were incubated in ammonium hydroxide for 30 min, then washed with water and incubated for 30 min in black and white film developer diluted 1:9 with water and then rinsed with double-distilled water. Grading concentrations of alcohol were used to dehydrate the brain slices which were then transferred into a CXA solution containing formyl trichloride, xylene, and absolute ethyl alcohol (1:1:1) for 15 min. Finally, slides were cover-slipped and visualized under the microscope (Nikon, Tokyo, Japan), whereby intact dendritic branches from the hippocampus were selected for spine counting.

### 2.5. Nissl Staining

The mice were anesthetized and then intracardially perfused with saline followed by 4% paraformaldehyde. The brains were harvested and postfixed for 24 h in the same 4% paraformaldehyde and dehydrated in 30% sucrose solution until the brains sink to the bottom then coronally sectioned at a thickness of 25 μm with a cryotome (CM1950, Leica, Nussloch, Strasse, Germany), and the free-floating sections were preserved in antifreeze solution (40% PBS, 30% glycerol, and 30% ethylene glycol) at −20 °C until used. Sections were mounted on gelatin-coated slides before Nissl staining was performed according to the manufacturer’s instructions (Beyotime Biotechnology, Shanghai, China). The slides were first incubated in Cresyl violet for 10 min at 25 °C, dehydrated through 50%, 75%, 95%, and 100% alcohol, then finally cleared in xylene and cover-slipped. A light microscope (Nikon, Tokyo, Japan) was used to snap the pictures, the Nissl-stained neurons in the hippocampal CA3 regions were counted, and the thickness of the cortex was measured.

### 2.6. Thioflavin S Staining

Thioflavin S staining (ThS) helps to stain beta sheets such as those present in Aβ plaques. Briefly, the antifreeze-preserved 25 μm brain slices were brought to room temperature and washed three times in PBS, and then stained with ThS (T1892-25G, Sigma-Aldrich, Shanghai, China) (12.5 mg/mL) in 50% ethanol in dark for 5–10 min at room temperature. This was followed by two washes with 50% ethanol and PBS then another 2 washes in PBS. The slides were then sealed using 50% glycerin in PBS. The Olympus VS120 S6 slide scanner system (Tokyo, Japan) was used to snap the images and the immunofluorescence intensity was evaluated using ImageJ software (1.51n, Wayne Rasband, Bethesda, MD, USA).

### 2.7. Western Blot Analysis

The mice were deeply anesthetized and perfused with normal saline before the brains were harvested and hippocampi were separated and homogenized in RIPA lysis buffer and the Western blot was performed as previously described [21]. Briefly, equal quantities of proteins were separated and transferred to a nitrocellulose membrane, blocked, and incubated overnight with primary antibodies (Table 1) at 4 °C, then with the secondary antibodies. The blots were visualized with the Odyssey (LICOR Biosciences, Boston, MA, USA) before ImageJ software was used to quantify the density of the bands. All blot gels are shown in Appendix A.

### 2.8. PP2B Activity Assay

PP2B activity was assayed using the Cellular Calcineurin Phosphatase Activity Assay Kit (Colorimetric) (ab139464) according to the manufacturer’s instructions (Abcam, Boston, MA, USA). Briefly, supernatants from mice brain tissue extracts were prepared with the provided lysis buffer. Endogenous-free phosphate was firstly removed from the supernatants then the background, total phosphatase activity, total phosphatase activity less PP2B, and the positive control wells were prepared in duplicates for each sample. To each well of the calcineurin samples except for the “background” control, a volume of 10 µL of the calcineurin substrate was added and equilibrated for 10 min at 30 °C. Following this, a volume of 5 µL extract or diluted calcineurin was added to appropriate wells and incubated at 30 °C for 30 min. Finally, the reaction was terminated by adding 100 µL of Green Assay Reagent, and the color was allowed to develop for 30 min. The amount of phosphate released from the substrate was detected by measuring the absorbance of a molybdate-malachite green-phosphate complex at 620 nm. The PP2B activity equals total phosphatase activity minus total phosphatase activity less PP2B.

### 2.9. ELISA Assay for Aβ40/42, IL-1β and TNF-α

The mice were deeply anesthetized, the brain removed, and the hippocampi were isolated then homogenized with PBS (containing 1:100 PMSF and 1:100 protease inhibitor cocktail) and centrifuged at 12,000× *g* at 4 °C for 10 min. The RIPA-soluble supernatant fraction was collected for detecting Aβ40/42, IL-1β, and TNF-α levels. Protein concentration in the supernatant was measured by the BCA method, and 200 μg of total protein in 100 μL PBS was added for the assay. The amount of Aβ40 and Aβ42 were detected in protein soluble fractions by using a sandwich ELISA kit according to the manufacturer’s instructions (Elabscience Biotechnology, Wuhan, China). The amount of IL-1β and TNF-α were detected in protein soluble fractions by using a sandwich ELISA kit according to the manufacturer’s instructions (ABclonal, Wuhan, China).

### 2.10. Statistical Analysis

The data represent the Mean ± SEM and were analyzed using GraphPad Prism8 (GraphPad Software Inc., San Diego, CA, USA). The difference between the two groups was assessed using an unpaired Student’s t-test, while that among more than two groups was assessed by one- or two-way analysis of variance or repeated measured analysis of variance followed by a post-hoc test. Statistical significance was set at *p* ≤ 0.05.

## 3. Results

### 3.1. Moringa oleifera Improves Behavioral and Cognitive Alterations in APP/PS1 Mice

Aβ is known to trigger the pathological changes that result in AD which clinically translates as cognitive impairment. MO was shown to have some anti-oxidative, anti-inflammatory, and neuroprotective properties all of which are implicated in AD. In this study, three-month-old APP/PS1 mice were treated with a 400 mg/kg/day dose of MO, following which we carried out a panel of behavioral experiments and found out that APP/PS1 mice exhibit memory and behavioral impairments which are abrogated by the MO (Figure 1).

Firstly, we performed Morris Water Maze and found out that compared with the WT control, the APP/PS1 mice showed a longer escape latency during the six training days whereas the escape latency remains similar to the WT in the APP/PS1 treated with MO (Figure 1A). Moreover, on the test day, a longer latency to cross the position of the platform, less crossing times of the platform position, and a lesser time spent in the target quadrant were observed in the APP/PS1 mice compared with the WT and the MO treatment groups (Figure 1B–E). No difference was observed in the distance covered among all three groups (Figure 1F) indicating normal motor functions. Next, the open field test was carried out (Figure 1G–J) and the results showed that both the time spent in the center and the distance traveled were increased in the APP/PS1 mice, while in the APP/PS1 mice treated with MO these remain comparable with the WT control. However, the movement time is similar in all groups. These results indicate anxiety-like behavior and restlessness in the APP/PS1 mice which were improved by MO. A novel object recognition test was also performed and the results revealed an increased time spent exploring the new object, as well as a higher recognition index in both the WT control and MO-treated APP/PS1 mice compared with the APP/PS1 mice (Figure 1K,L). Lastly, a fear conditioning test was performed, and the results of the contextual fear memory test showed lower freezing episodes and freezing time and a higher freezing latency in the APP/PS1 mice when compared with the WT and the MO-treated APP/PS1 mice (Figure 1M–O). The cued fear memory results were not different among all groups (Appendix A). Together, these results indicate behavioral and cognitive deficits in the APP/PS1 mice which were significantly abrogated by the MO treatment, implying the role of MO in synaptic plasticity.

### 3.2. Moringa oleifera Alleviates Aβ Level and Plaques Burdens in APP/PS1 Mice

Aβ can induce synaptic dysfunction [63] and thus lead to behavioral deficits. To investigate how MO improved cognitive functions in these mice, we first evaluated the Aβ levels (Figure 2). The results from the Western blot from the brain hippocampal lysates revealed an increase in the total APP protein level in the APP/PS1 mice compared with the WT with no significant difference between APP/PS1 with or without MO treatment (Figure 2A,B). However, compared with APP/PS1 without MO treatment, the APPβ level was significantly decreased in APP/PS1 mice with MO treatment and this is comparable with the WT control (Figure 2A,C). Moreover, the thioflavin S staining results show similar patterns to the Western blot. Significantly higher plaque areas were observed in both the cortex and the hippocampus of the APP/PS1 mice compared with the WT and these were abrogated by MO treatment in the APP/PS1 mice (Figure 2D–F). Furthermore, to confirm our results, we evaluated both Aβ40 and Aβ42 via ELISA. Both Aβ40 (Figure 2G) and Aβ42 (Figure 2H) were found to be significantly higher in the APP/PS1 mice compared with the WT, and MO treatment reduces it to WT control levels. These results together indicate that MO treatment can significantly improve the Aβ load in APP/PS1 mice which might be at least in part responsible for the improved behavioral and cognition functions in these mice.

### 3.3. Moringa oleifera Modulates Both Production and Clearance Pathways of Aβ in APP/PS1 Mice

The Aβ peptides result from the successive cleavage of APP by BACE1 and the γ-secretase [3,4]; thus, we evaluated the levels of these proteins and the result showed that in the APP/PS1 mice MO downregulates BACE1, the rate-limiting enzyme in Aβ production, to a level comparable with the WT control, but the level of PS1, the catalytic subunit of γ-secretase, remains unchanged (Figure 3A–C). It was reported that Asparagine endopeptidase (AEP), a pH-controlled cysteine proteinase, can cleave APP at N585, BACE1 at N294, and Tau at N368 residues to mediate AD [64,65,66,67,68,69]. Therefore, we measured the AEP level in the APP/PS1 following the MO treatment. Our results show an upregulation of AEP in the APP/PS1 mice compared with WT mice whereas following MO treatment, the AEP level was significantly reduced although it is higher than in the WT control mice (Figure 3A,D). Aβ clearance-associated proteins including insulin-degrading enzyme (IDE), neprilysin (NEP), and the low-density lipoprotein receptor-related protein 1 (LRP1) were also evaluated. Interestingly, our results revealed a decrease in these proteins in the APP/PS1 mice compared with the WT ones, and the MO supplement significantly improved these proteins to the WT control level (Figure 3A,E–G).

### 3.4. Moringa oleifera Improves p-Y1472 GluN2B by Decreasing STEP in APP/PS1 Mice

Synaptic dysfunction is one of the characteristics of AD and translates to behavioral and cognitive deficits observed in different stages of AD [28] and Aβ oligomers aggregation process negatively affects synaptic structure and function of neuronal networks and synaptic plasticity [70,71,72,73]. NMDARs play a crucial role in synaptic plasticity and thus learning and memory [30,74,75]; thus, we evaluated the expression of NMDAR subunits. We found that the protein levels of GluN1, GluN2A, and GluN2B were not different (Figure 4A–D); however, the phosphorylated GluN2B at Tyr1472 (p-Y1472 GluN2B) was found to be significantly downregulated in the APP/PS1 mice compared with the WT, and MO supplement rescued this alteration (Figure 4A,E). To understand how the p-Y1472 GluN2B is decreased, we evaluated the phosphatase and kinase, STEP, and Fyn, that regulate the phosphorylation of GluN2B at this site. Our results showed an upregulation in both total and non-phosphorylated (active) STEP at Ser221 (np-S221 STEP) in the APP/PS1 mice compared with the WT mice and this was abolished by MO treatment (Figure 4F–H). Surprisingly, we found that the level of total Fyn, the main kinase of GluN2B, was significantly increased in the APP/PS1 mice compared with the WT (Figure 4F,I), but the phosphorylation level of Fyn at Tyr416 (p-Y416 Fyn) was tremendously reduced compared with the WT mice (Figure 4F,J). Interestingly, both the increase in the total and the decrease in the p-Y416 Fyn were recovered by the MO treatment. These data together suggest that APP/PS1 mice exhibit synaptic plasticity dysfunction seen as decreased phosphorylation of NMDA receptors due to the upregulation of both the level and activity of STEP.

### 3.5. Moringa oleifera Modulates the PP2B/DARPP-32/PP1 Axis to Decrease STEP in APP/PS1 Mice

We found that STEP is increased in the APP/PS1 mice and Aβ was reported to induce intracellular calcium influx and lead to increased STEP activity via the PP2B/DARPP-32/PP1 axis [40,41,42]. Therefore, in this study, we evaluated this axis. Our data showed that the total levels of both DARPP-32 and PP1 were unchanged among all groups, but their phosphorylation levels at Thr34 (p-T34 DARPP-32) and Thr320 (p-T320 PP1) were significantly decreased in the APP/PS1 mice compared with the WT mice (Figure 5A–E). Interestingly, the treatment with MO abolished these changes. Furthermore, we evaluated the activity of the upstream phosphatase PP2B that dephosphorylates DARPP-32, and the result revealed an increase in the activity of this enzyme in the APP/PS1 mice compared with the WT and the MO-treated APP/PS1 mice (Figure 5F). These results suggest that MO modulates the PP2B/DARPP-32/PP1 axis to downregulate STEP activity thereby improving GluN2B Tyr1472 phosphorylation in APP/PS1 mice.

### 3.6. Moringa oleifera Improves Synaptic Loss and Neurodegeneration in APP/PS1 Mice

Synaptic function is dependent on the structural integrity of the synapse which is mediated by both pre-synaptic and post-synaptic elements such as synapsin1 and PSD95. These synaptic proteins, together with NMDARs are critical for synaptic plasticity [30], thus we evaluated their levels through Western blot. Our results showed that APP/PS1 mice exhibit a significant decrease in the levels of PSD95 and synapsin1 that were improved to WT mice levels in the APP/PS1 mice treated with MO (Figure 6A–C). To evaluate synaptic structures, we performed Golgi staining and the results revealed a decrease in both the total and mushroom-type spines in the APP/PS1 mice that were maintained at WT mice level following treatment with MO (Figure 6D–F). Next, we examined neuronal integrity by Nissl staining and found significantly decreased neurons in the CA3 hippocampal region in the APP/PS1 mice that were improved to WT mice level in the APP/PS1 group treated with MO (Figure 6G,H). Moreover, APP/PS1 mice exhibit a decrease in cortical thickness that was also abrogated by MO treatment (Figure 6G,I). These data together suggest that MO treatment can improve synaptic alterations and neuronal loss that occur in APP/PS1 mice.

### 3.7. Moringa oleifera Improves Neuroinflammation in APP/PS1 Mice

More and more shreds of evidence are now clear that neuroinflammation is a significant player in the pathogenesis of neurodegenerative diseases including AD. The major inflammatory cells in the CNS include microglia and astrocytes and together with cytokines such as TNF-α and IL-1β are critical to AD pathogenesis [76,77,78,79,80,81]. We, therefore, measured the level of microglia (IBA1) and astrocyte (GFAP) markers, TNF-α, and IL-1β. The data revealed that all evaluated proteins were significantly upregulated in the APP/PS1 mice compared with the WT and the MO-treated APP/PS1 mice groups (Figure 7A–E). These results confirm that neuroinflammatory processes occur in the APP/PS1 mice and that MO can reduce these inflammatory markers to levels comparable with WT control mice.

## 4. Discussion

The burden of Alzheimer’s disease is increasing globally due to the increase in the elderly population. Aβ initiates many cascades that result in Tau pathology, neuroinflammation, neurodegeneration, and synapse loss which clinically manifest as cognitive deficits [81,82,83]. Despite decades of research, there is no cure for AD. In this study, we report that a four-month MO treatment significantly alleviates AD-like pathologies in APP/PS1 mice thereby improving cognitive and behavioral deficits.

Increased production of Aβ was reported in the early stage of AD even before overt toxicity [84,85] and this was shown to be associated with synaptic alterations and cognition deficits [86,87]. APP/PS1 mice are an AD model that overproduces Aβ; therefore, we evaluated the Aβ load in these mice following MO treatment. Our results revealed an increased Aβ load in the APP/PS1 mice that was tremendously alleviated by the MO treatment as indicated by the western blot, thioflavin S, and ELISA experiments. This is consistent with our previous study [21] and a recent study that showed that the component of MO Niazimicin decreases Aβ in albino rate [27], confirming the ability of MO to reduce Aβ burden. 

The level of Aβ is a function of the production and clearance systems. Our results revealed a decrease in BACE1, the Aβ production rate-limiting enzyme, in the APP/PS1 mice treated with MO to a level comparable with the WT control mice. BACE1 was reported to be associated with a cascade in which oxidative stress can induce intracellular calcium influx that activates calpain which results in the activation of BACE1 transcription factors STAT3 and NFAT1 [88]. MO is known to have strong antioxidant properties and this might explain the reduction in the BACE1 level in the MO-treated mice. Increasing evidence shows that AEP, also known as delta secretase, is involved in the amyloidogenic processing of the APP [64,66,67]. APP 586–695, a product of APP cleavage by AEP, was found to increase the expression of AD-related genes and AD pathogenesis [66]. Clearance of APP 586–695 in 5xFAD mice or blockage of AEP truncation of APP ameliorates Aβ pathology and cognitive impairments [66]. Moreover, AEP can not only cleave Tau to produce Tau 1–368, which activates STAT1 and increases the BACE1 production [65] but also can cleave BACE1 at N294 and increase its protease activity [64], thereby upregulating Aβ production. Interestingly, AEP can be activated by oxidative stress [89,90] and oxidative stress is reported in APP/PS1 mice [91,92]. Our results revealed an increased level of AEP in the APP/PS1 that was significantly abrogated by MO treatment. These results suggest that AEP and BACE1 might crosstalk to increase Aβ production and that MO ameliorates these alterations possibly via its antioxidative potential.

NEP [93,94] and IDE [95] are some of the critical players in the extracellular and intracellular degradation of Aβ and thus its clearance. IDE regulates the Aβ level in vivo [96] and is found in most cellular compartments and can degrade β-structure forming peptides that are associated with neurodegeneration [97]. IDE is reported to decrease with age and in the early stage of AD [95], and genetic alterations associated with a higher risk of AD were reported in IDE and NEP genes [98,99]. Another protein associated with Aβ clearance is LRP1 which is normally reduced with increasing age and is further decreased in AD [100,101,102]. Interestingly, studies have revealed that both pharmacologic and genetic inhibition of LRP1 resulted in the buildup of Aβ42 [103,104]. In line with these studies, we found that IDE, NEP, and LRP1 were significantly reduced in APP/PS1 mice while MO treatment helped to prevent these losses. Altogether, these data suggest that MO downregulates the amyloidogenic processing of APP as well as improves Aβ clearance to decrease the Aβ burden in these mice.

Synaptic plasticity is the basic process via which new memories are formed and NMDARs are critical for synaptic plasticity and thus learning and memory [30,74,75]. The GluN2B subunit of NMDAR is the most significantly tyrosine-phosphorylated protein in the PSDs [105], and its synaptic expression is significantly regulated by its phosphorylation [106]. Moreover, the activity-dependent subunit-specific phosphorylation of NMDARs significantly impacts their synaptic localization and function [107,108]. Here, we observed a decrease in p-Y1472 GluN2B in the APP/PS1 mice, and the MO treatment rescued it to the WT control level. In addition, consistent with previous reports [37,109,110,111], our data revealed an increase in the total level of STEP as well as the np-S221 STEP. In line with this, a decrease in the p-Y416 Fyn was observed; however, unexpectedly an increase in the level of total Fyn was observed in the APP/PS1 mice, possibly indicating a feedback mechanism from the mouse system to compensate for the decreased level of the active p-Y416 Fyn. Alterations in the PP2B/DARPP-32/PP1 axis including increased activities of PP1, the major STEP phosphatase, and PP2B, a Ca^2+^/calmodulin-dependent phosphatase that can be activated by oxidative stress-mediated intracellular calcium influx, were observed in the APP/PS1 mice. These changes were prevented by MO in the treated mice. Interestingly, calcium overload has been reported in the brain of APP/PS1 mice with plaques [70,71], and increased Aβ plaques which were ameliorated by MO were observed in our study. These together indicate that MO prevented the loss of p-Y1472 GluN2B by downregulating STEP which can directly or indirectly (via decreasing p-Y416 Fyn) decrease the GluN2B phosphorylation. In turn, the decreased STEP activity might at least in part be due to the MO-mediated decrease in oxidative stress resulting in decreased calcium influx and PP2B activity downregulation.

APP/PS1 mice are reported to exhibit significant downregulation of synaptic proteins such as PSD95 and synapsin1 [62,112,113]. In line with these studies, the protein levels of PSD95 and synapsin1 were found to be downregulated in the APP/S1 mice. Interestingly, consistent with our previous study [21], MO significantly recovered these alterations in the treated mice. The scaffolding protein and PSD organizer PSD95 is the major component of PSD and is central to the glutamatergic synaptic signaling [114]. PSD95 is a key player in synaptic transmission and LTP by favoring the surface expression of GluN2B-containing NMDARs via direct interaction and by inducing STEP’s ubiquitination and degradation [115,116,117,118]. Moreover, it was recently demonstrated that PSD95 can protect synapses against Aβ toxicity [119], suggesting that a decrease in synaptic PSD95 is an indication of synaptic vulnerability to Aβ in AD. Interestingly, the downregulation of neuronal PSD95 is believed to be mediated by calpain [120], a calcium-dependent protease that can be activated under oxidative stress conditions [88]. Synapsin1 is known to play roles in neurogenesis, synapse formation, and synaptic transmission [121]. These together further suggest the role of MO in maintaining normal synaptic function via upregulating the level of PSD95 and synapsin1.

The loss of dendritic synaptic plasticity is a key neurobiological basis of dementia and occurs in the brains of AD patients [122,123]. Dendritic spines are key players in synaptic plasticity mainly because they contain post-synaptic elements and have an inherent ability to undergo dynamic changes in response to incoming synapses stimuli [124,125]. In our study, we observed a decrease in the total number and the mushroom-type spines in APP/PS1 mice compared with the WT control, and these changes were prevented by the MO treatment. NMDARs are implicated in the structural regulation of spines, suggesting that the MO-mediated STEP downregulation which helps maintain the surface expression of GluN2B might contribute to the improved dendritic spine density in the MO-treated APP/PS1 mice. Moreover, PSD95 was reported to regulate dendritic arborization and spine number by interacting with cysteine-rich PDZ-binding protein and neuronal cytoskeleton [126]. Interestingly, the decreased PSD95 in APP/PS1 mice was rescued by MO treatment further implying the role of MO in maintaining dendritic spine density. Recently, results from a study suggest that mushroom-type spines are more liable to respond to dynamic changes during synaptic transmission, and their content correlates more with synaptic strength [127]. This further suggests that MO maintains synaptic strength and normal functions by preserving mushroom-type spines. Nissl staining data indicate a decrease in the number of neurons in the CA3 region of the hippocampus and a decreased cortical thickness in the APP/PS1 mice, suggesting neurodegeneration. This was rescued in the MO-treated mice. It was shown that acute depletion of PSD95 leads to hippocampal neuron death [128], and neurons lacking PSD95 exhibit neuronal damage as a result of NMDARs-mediated excitotoxicity [129]. This suggests that upregulating PSD95 might play a role in the MO-mediated improvement of neurodegeneration and this is in line with our previous data [21]. Moreover, a recent study also showed that Niazimicin, one of MO’s active components, has a neuroprotective effect via decreasing malondialdehyde, cholinesterase, nitric oxide, Aβ, caspase-3, and inducible nitric oxide synthase enzymes [27]. Other components of MO such as ferulic acid, quercetin, and linalool have been demonstrated to exert neuroprotective effects [130,131,132].

Apart from Aβ, Tau, and synapses loss, neuroinflammation is another important player in the pathogenesis of AD [133,134,135,136] and MO has been proven to possess anti-inflammatory effects [137,138]. Consistent with this, we found that treatment with MO in the APP/PS1 mice resulted in a significant decrease in the GFAP and IBA1 markers of astrocytes and microglia activation, as well as the cytokines TNF-α and IL-1β. The interrelation of Aβ, Tau, and glial cells in space and time in the brain results in the Aβ induction of neuroinflammation which then influences the generation of the Aβ [139]. Clear evidence highlights that TNF-α is increased and centrally involved in the AD [76,77,78], while IL-1β inhibits the hippocampal LTP [140]. Moreover, TNF-α functions as a gliotransmitter that regulates synaptic function and strength and directly affects the glutamate transmission [77,141,142]. The component of MO, ferulic acid, was reported to inhibit Aβ-induced microglial activation in mice [143]. Therefore, by improving neuroinflammation MO might play a role in the management of AD.

Aβ induces behavioral deficits, including hyperactivity, impaired new object recognition, spatial working, and reference memories [143,144,145]. Synaptic plasticity and synapse integrity are critical for learning and memory [30,74,75]. In this study, MO was found to improve Aβ load, synaptic proteins, neurodegeneration, and neuroinflammation which together might explain the amelioration of behavioral, cognitive, learning, and memory impairments in APP/PS1 mice treated with MO. This is in line with previous reports where MO extract or components in MO were demonstrated to improve cognitive and behavioral impairments [21,57,130].

It should be noted that our experiment presents some limitations which include: (i) the use of male animals only in the experiment; (ii) no MO-treated WT animal group to explore the effect of MO in these animals; and (iii) no vehicle (methanol) control treatment in both the WT and APP/PS1 animals. These deficiencies will be considered in our future studies for better reliability of data, more originality, and standardization of experimental research.

## 5. Conclusions

MO is a compound with multiple biological functions. We here report that treatment with MO prevented the increase in Aβ load in APP/PS1 mice via decreasing the enzymes that promote Aβ production including BACE1 and AEP, as well as increasing the proteins responsible for Aβ clearance including IDE, NEP, and LRP1. Moreover, MO maintained the phosphorylation status of GluN2B via decreasing STEP activity, improved the level of synaptic plasticity-related proteins such as PSD95 and synapsin1, and prevented dendritic spine loss and neurodegeneration. These together result in improved behavioral and cognitive deficits. The data from our study provides insight into the use of MO as a nutraceutical agent in the management of AD.

## Figures and Tables

**Figure 1 nutrients-14-04284-f001:**
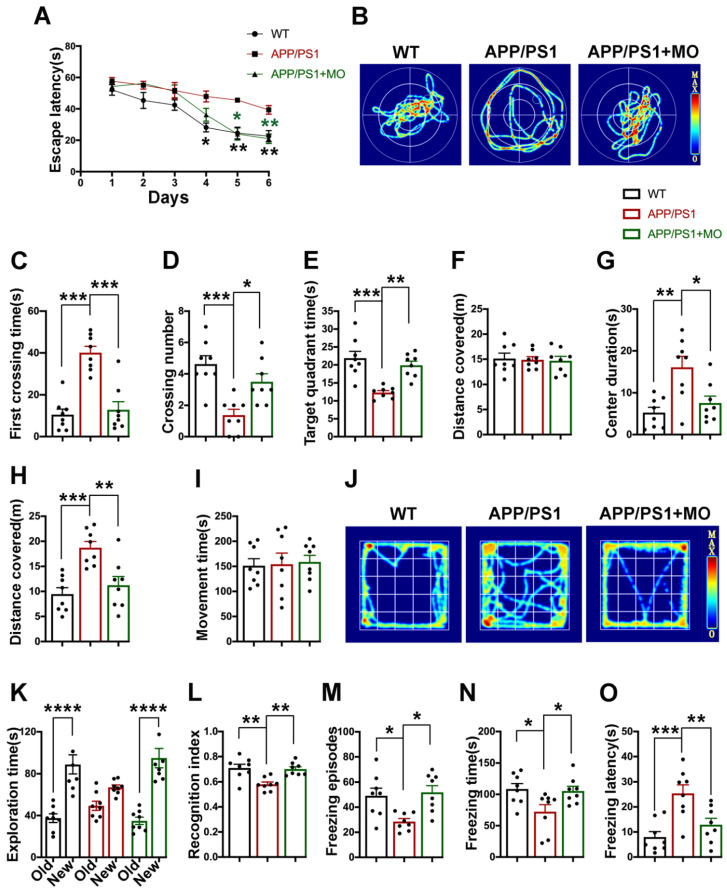
Long-term MO treatment prevents behavioral and cognitive deficits in APP/PS1 mice. (**A**–**F**) Morris Water Maze results. (**A**) The escape latency (s) of mice to find the platform during the six days of training; (**B**) the representative traces of mice searching the target during the probe test; (**C**) the first crossing time (s); (**D**) crossing number; (**E**) the time (s) spent in the target quadrant; and (**F**) the total distance (m) covered by the mice during the probe test. (**G**–**J**) Open field test results. (**G**) Center duration (s); (**H**) the distance (m) covered; (**I**) the movement time (s); and (**J**) the representative movement traces of mice during the five minutes of the test. (**K**,**L**) Novel object recognition test results. (**K**) The exploration time (s) of old and new objects during the test; and (**L**) the recognition index. (**M**–**O**) Fear conditioning test results. (**M**) The freezing episodes; (**N**) the freezing time (s); and (**O**) the freezing latency (s) during the test time. Data are presented as Mean ± SEM, (*n* = 8) for each group. * *p* < 0.05; ** *p* < 0.01; *** *p* < 0.001; and **** *p* < 0.0001 vs. WT or vs. APP/PS1 mice groups.

**Figure 2 nutrients-14-04284-f002:**
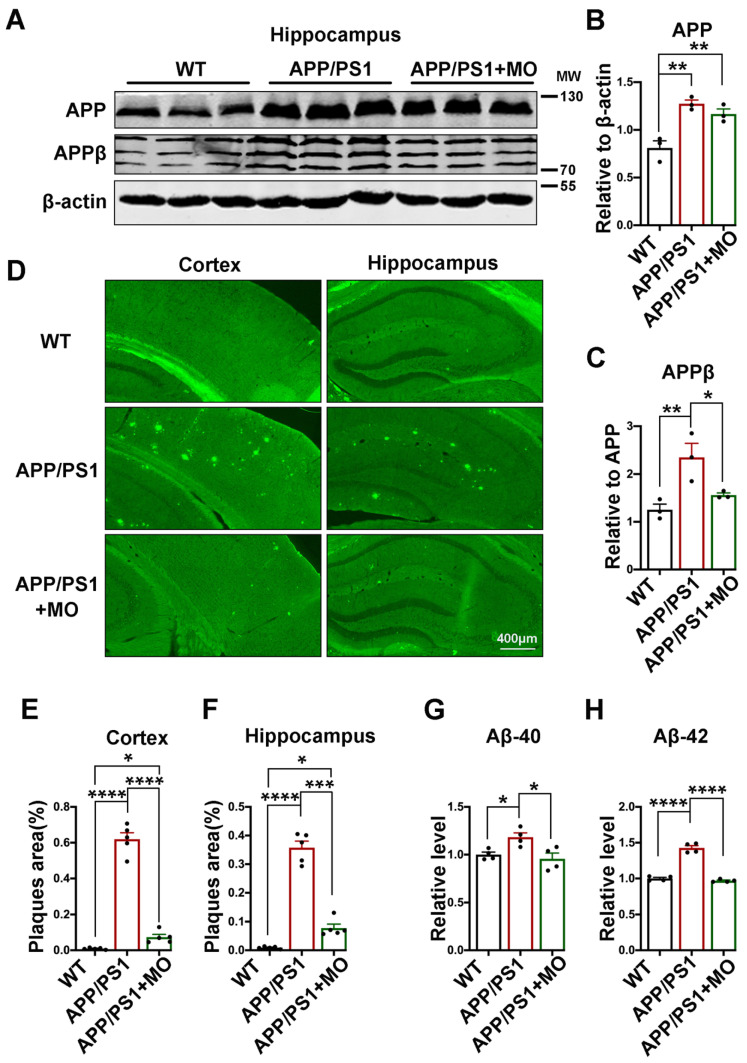
Long-term MO treatment alleviates the Aβ burden in APP/PS1 mice. (**A**) The expression of APP and APPβ from hippocampal lysates of WT control and APP/PS1 mice with or without MO treatment was evaluated by Western blotting, (*n* = 3). β-actin serves as the loading control. (**B**,**C**) Statistical analysis of APP and APPβ. (**D**) Representative images of thioflavin S staining from WT control and APP/PS1 mice with or without MO treatment (scale bare = 400 μm, *n* = 3). (**E**,**F**) The quantification of thioflavin S fluorescence of the % areas occupied by the Aβ plaques in the cortex and hippocampus. (**G**,**H**) The statistical analysis of the ELISA test for Aβ40 and Aβ42 from hippocampal lysates of WT control and APP/PS1 mice with or without MO treatment (*n* = 4). The data are presented as Mean ± SEM. * *p* < 0.05; ** *p* < 0.01; *** *p* < 0.001; and **** *p* < 0.0001 vs. WT or vs. APP/PS1 mice groups.

**Figure 3 nutrients-14-04284-f003:**
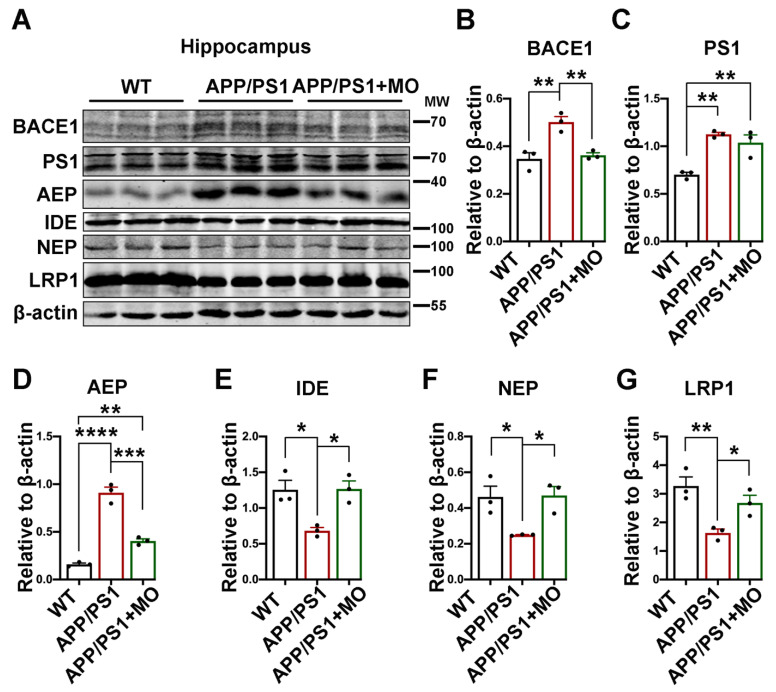
MO decreases production and increases clearance of Aβ in APP/PS1 mice. (**A**) The expression level of BACE1, PS1, AEP, IDE, NEP, and LRP1 from hippocampal lysates of WT mice and APP/PS1 mice with or without MO treatment was evaluated by Western blotting. (**B**–**G**) The quantification of the Western blot bands from panel A. β-actin serves as the loading control. The data are presented as Mean ± SEM, (*n* = 3). * *p* < 0.05; ** *p* < 0.01; *** *p* < 0.001; and **** *p* < 0.0001 vs. WT or vs. APP/PS1 mice groups.

**Figure 4 nutrients-14-04284-f004:**
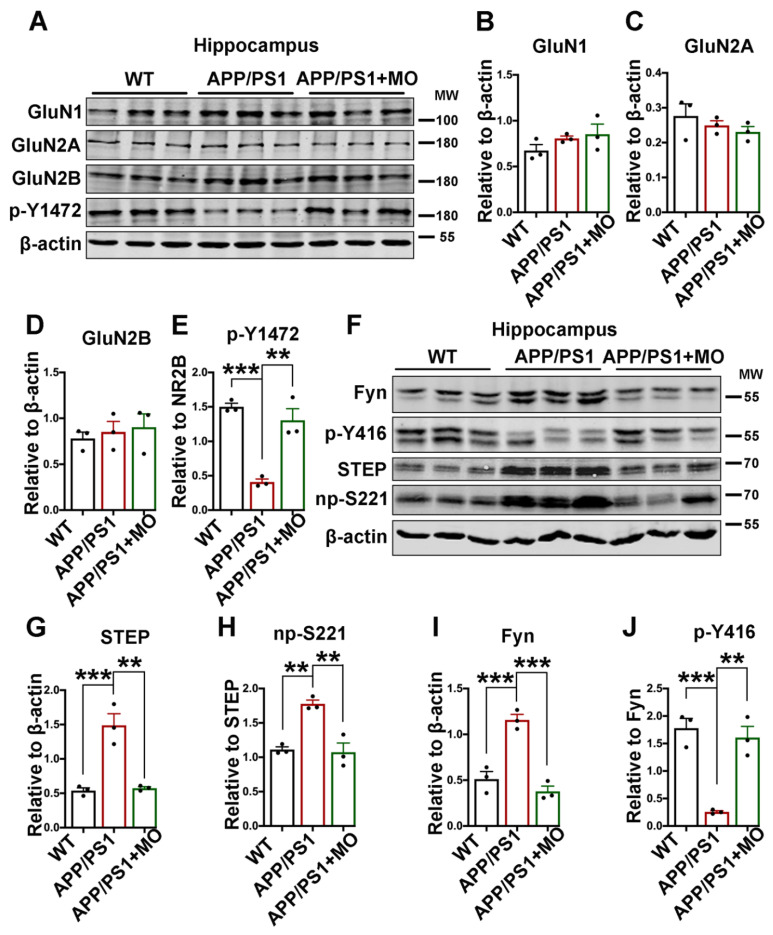
MO improves GluN2B phosphorylation by decreasing STEP in APP/PS1 mice. (**A**) The expression level of GluN1, GluN2A, GluN2B, and p-Y1472 from hippocampal lysates of WT and APP/PS1 mice with or without MO treatment was evaluated by Western blotting. (**B**–**E**) The statistical analysis of Western blot bands from panel A. (**F**) The expression level of STEP, np-S221, Fyn, and p-Y416 from hippocampal lysates of WT control and APP/PS1 mice with or without MO treatment was examined. (**G**–**J**) The statistical analysis of Western blot bands from panel F. β-actin serves as the loading control. The data are presented as Mean ± SEM, (*n* = 3). ** *p* < 0.01 and *** *p* < 0.001 vs. WT or vs. APP/PS1 mice groups. p-Y1472 = phosphorylated GluN2B at Tyr1472; np-S221 = non-phosphorylated STEP at Ser221; p-Y416 = phosphorylated Fyn at Tyr416.

**Figure 5 nutrients-14-04284-f005:**
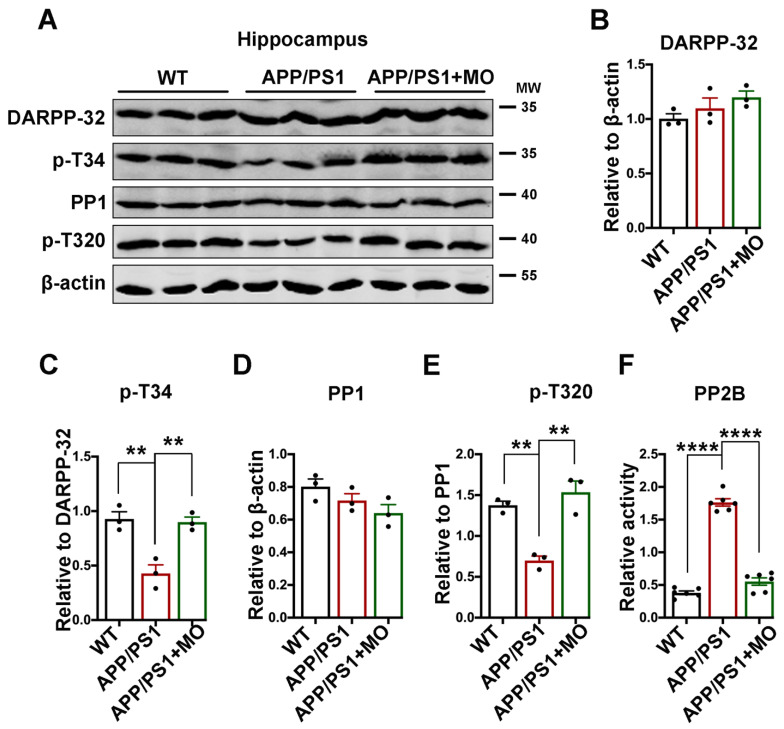
MO downregulates STEP by modulating the PP2B/DARPP-32/PP1 axis. (**A**) The expression level of DARPP-32, p-T34, PP1, and p-T320 from hippocampal lysates of WT control and APP/PS1 mice with or without MO treatment was evaluated by Western blotting, (*n* = 3). (**B**–**E**) The statistical analysis of Western blot bands from panel A. β-actin serves as the loading control. (**F**) The statistical analysis of the PP2B activity test from hippocampal lysates of WT control and APP/PS1 mice with or without MO treatment, (*n* = 4). The test was carried out using a PP2B activity kit. The data are presented as Mean ± SEM. ** *p* < 0.01 and **** *p* < 0.0001 vs. WT or vs. APP/PS1 mice groups. p-T34 = phosphorylated DARPP-32 at Thr34; p-T320 = phosphorylated PP1 at Thr320.

**Figure 6 nutrients-14-04284-f006:**
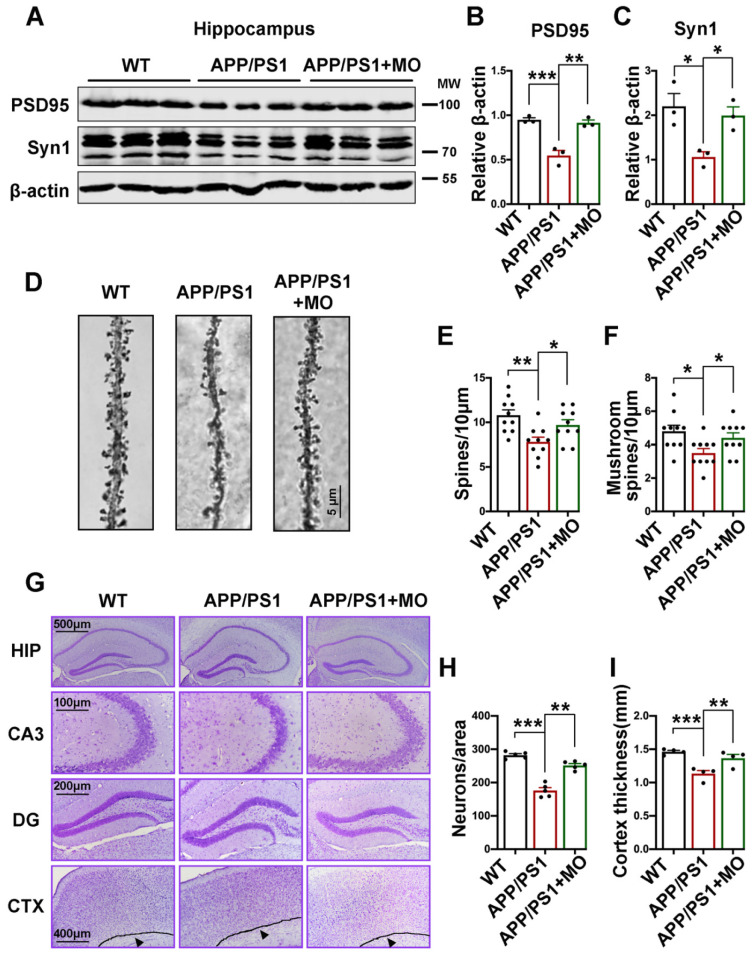
Long-term MO treatment prevents synaptic proteins and dendritic spines losses and neurodegeneration in APP/PS1 mice. (**A**) The expression level of PSD95 and Synapsin1 from hippocampal lysates of WT control and APP/PS1 mice with or without MO treatment was examined by Western blotting, (*n* = 3). (**B**,**C**) The statistical analysis of Western blot bands from panel A. β-actin serves as the loading control. (**D**–**F**) Golgi staining results. (**D**) The representative micrographs of the Golgi staining experiment (scale bare = 5 μm, *n* = 3 per group, 3–4 dendrites per mouse). (**E**) Total spines number per 10 μm area. (**F**) The mushroom-type spines per 10 μm area. (**G**–**I**) Nissl staining results (scale bars = 100, 200, 400, and 500 μm, *n* = 3). (**G**) The representative micrographs of the Nissl staining experiment. (**H**) The number of neurons per area. (**I**) The thickness (mm) of the cortex. The data are presented as Mean ± SEM. * *p* < 0.05; ** *p* < 0.01; and *** *p* < 0.001 vs. WT or vs. APP/PS1 mice groups.

**Figure 7 nutrients-14-04284-f007:**
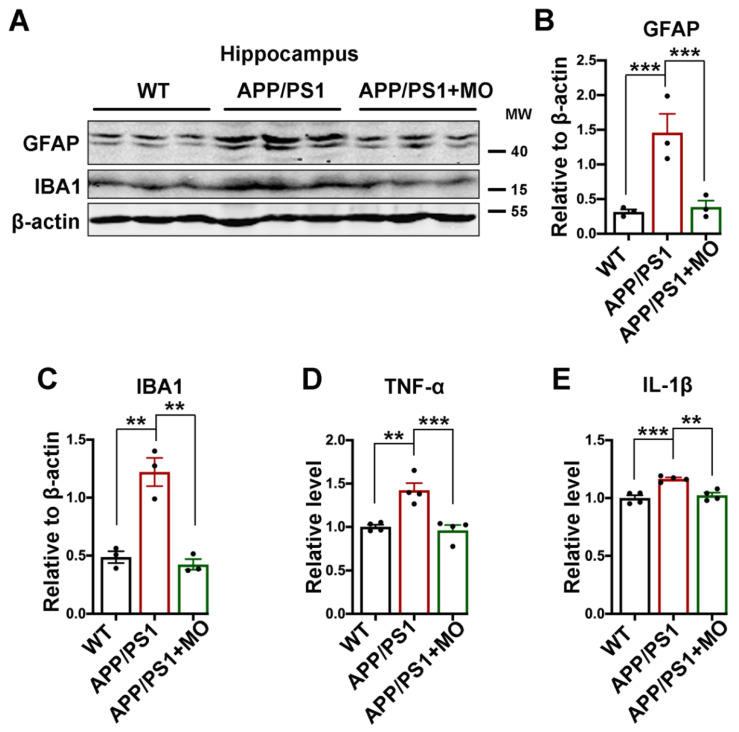
Long-term MO treatment alleviates neuroinflammation in APP/PS1 mice. (**A**) The expression level of GFAP and IBA1 from hippocampal lysates of WT control and APP/PS1 mice with or without MO treatment was evaluated by Western blotting, (*n* = 3). (**B**,**C**) The statistical analysis of Western blot bands from panel A. β-actin serves as the loading control. (**D**,**E**) The statistical analysis of TNF-α and IL-1β ELISA results from hippocampal lysates of WT control and APP/PS1 mice with or without MO treatment was evaluated, (*n* = 4). The data are presented as Mean ± SEM. ** *p* < 0.01 and *** *p* < 0.001 vs. WT or vs. APP/PS1 mice groups.

**Table 1 nutrients-14-04284-t001:** The primary antibodies used in this study.

Antibody	Specificity	Type	Species	Source (Catalog Number)
Anti-GluN1	GluN1	pAb	Rabbit	ABclonal (A7677)
Anti-GluN2A	GluN2A	mAb	Rabbit	ABclonal (A19089)
Anti-GluN2B	GluN2B C-terminus	pAb	Rabbit	ABclonal (A3056)
Anti-p-Y1472	p-GluN2B (Y1472)	pAb	Rabbit	Abcam (ab3856)
Anti-STEP	STEP (23E5)	pAb	Mouse	Cell Signaling Technology (4396)
Anti-np-S221	np-STEP (S221)	mAb	Rabbit	Cell Signaling Technology (5659)
Anti-FYN	FYN	mAb	Mouse	ABclonal (A0086)
Anti-p-Y416	p-Sar family Y416	pAb	Rabbit	ABclonal (RK06002)
Anti-PP1CA	PP1CA (a.a 1–330)	pAb	Rabbit	ABclonal (A12468)
Anti-p-T320	p-PP1CA T320	pAb	Rabbit	ABclonal (AP0786)
Anti-DARPP-32	DARPP-32	pAb	Rabbit	Abmart (Q9UD71)
Anti-p-T34	p-T34 DARPP-32	pAb	Rabbit	Abcam Ab254063
Anti-APP	APP (APP695, APP770, APP751)	pAb	Rabbit	Cell Signaling Technology (2452)
Anti-APPβ	sAPPβ	pAb	Rabbit	IBL (18957)
Anti-BACE1	BACE1 (D10E5)	mAb	Rabbit	Cell Signaling Technology (5606)
Anti-PS1	PS1	pAb	Rabbit	Cell Signaling Technology (3622)
Anti-AEP	Legumain (D6S4H)	mAb	Rabbit	Cell Signaling Technology (93627)
Anti-IDE	IDE	pAb	Rabbit	Abcam (ab32216)
Anti-NEP	CD10/MME	pAb	Rabbit	ABclonal (A5664)
Anti-LRP1	LRP1	pAb	Rabbit	ABclonal (A13509)
Anti-PSD95	PSD95 N-terminal	mAb	Rabbit	Cell Signaling (2507)
Anti-Synapsin1	Synapsin1	pAb	Rabbit	Millipore (AB1543)
Anti-GFAP	GFAP C-terminus	pAb	Rabbit	ABclonal (A14673)
Anti-IBA1	AIF1/IBA1	mAb	Rabbit	ABclonal (A19776)
Anti-β-actin	β-actin	pAb	Rabbit	ABclonal, China (AC026)

Abbreviations: p-: phosphorylated; np-: non phosphorylated, pAb: polyclonal antibody; mAb: monoclonal antibody.

## Data Availability

All data used in this study are available from the corresponding authors upon reasonable request.

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
