# Peer review of "Moringa Oleifera Alleviates Aβ Burden and Improves Synaptic Plasticity and Cognitive Impairments in APP/PS1 Mice"

_nutrients, 2022, doi:10.3390/nu14204284_

Round 1

Reviewer 1 Report

One of the drawbacks of this paper is that the description is too long than necessary; for example, 1) the first paragraph of Discussion is overlapped with Introduction, 2) the second and third paragraphs of Discussion is overlapped with Results, with the following paragraphs showing the same mistakes, 3) the methodology describes too details, so shortened at least to two thirds by citing the previous paper, and 4) the Results should describe only pure results without any interpretation or explanations.

As the authors adopted so many molecular players (Table 1) as targets, they have lost their orientation of writing especially at Discussion. So, the readers feel much flustration during reading and interpreting data, as the paper looks like a dictionary. The data themselves appear beautiful. For example, the authors are recommended to rewrite the Discussion by dividing four effects of MO that was described in Abstract. 

1)    Anti-oxidation property of MO

2)    Anti-inflammatory property of MO

3)    Anti-cholineesterase peoperty of MO

4)    Neuroprotective property of MO

Alernatively, if Discussion is arranged according to the style (stream) of Conclusions, the readers can easily understand the value of the preset results by summarizing thoughts.

Author Response

  1. One of the drawbacks of this paper is that the description is too long than necessary; for example, 1) the first paragraph of Discussion is overlapped with Introduction, 2) the second and third paragraphs of Discussion is overlapped with Results, with the following paragraphs showing the same mistakes, 3) the methodology describes too details, so shortened at least to two thirds by citing the previous paper, and 4) the Results should describe only pure results without any interpretation or explanations.

Answer: Thank you very much for these suggestions. Following the constructive suggestions of the reviewer, we have now deleted most of these mentioned paragraphs and restructured them. We have also shortened some parts of the methodology and results. Please see the corresponding sections in our revised manuscript.

  1. As the authors adopted so many molecular players (Table 1) as targets, they have lost their orientation of writing especially at Discussion. So, the readers feel much flustration during reading and interpreting data, as the paper looks like a dictionary. The data themselves appear beautiful. For example, the authors are recommended to rewrite the Discussion by dividing four effects of MO that was described in Abstract. 

         1)    Anti-oxidation property of MO

         2)    Anti-inflammatory property of MO

         3)    Anti-cholineesterase peoperty of MO

         4)    Neuroprotective property of MO

Alernatively, if Discussion is arranged according to the style (stream) of Conclusions, the readers can easily understand the value of the preset results by summarizing thoughts.

Answer: We are very grateful for this very constructive suggestion from the reviewer. We have now summarized and rewritten the discussion in the flow style of the conclusions. Please see the discussion section in our revised manuscript.

Reviewer 2 Report

The authors submitted an interesting experimental study about the neuroprotective effects of an extract of the plant Moringa Oleifera (MO) in APP/PS1 mice.

The study includes a wide number of techniques that as a whole demonstrated the multiple mechanisms activated by MO. All determinations are well done and results are interesting. However the manuscript should be edited to improve clarity and concretion.

1 – In the abstract, C57 (line 28, and also all over the text and figures) is not an accepted abbreviation and the full strain name should be used instead. Otherwise, WT abbreviation is introduced in line 95.

2 - In Introduction, relevance of amyloid peptide generation for AD is addressed (lines 54 ss). However, please comment that there is a consensus in sporadic AD that an age-related deficiency of amyloid clearance has also a role in amyloid beta accumulation. Therefore inhibiting its generation may only partially contribute to an improvement.

3- The phosphorylation sites analyzed from the diverse proteins are not introduced (only Y1472, line 76). Please add briefly to show relevance although there were no previous results in TgAD animals. That can then be reduced in Discussion.

4- There are some points to improve in the experimental design that the authors should consider in future work: (i) The use of both sexes is increasingly required in AD experimental research; (ii) It is recommended to add a WT treated group, please Indicate the known effects of MO treatment in the WT animals if available; (iii) Control WT and APP/PS1 must receive the vehicle (methanol) used in the MO-treated APP/PS1 animals. Please comment these deficiencies in limitations of the study.

 5- In Results, the name of phosphorylated proteins in graphs titles, legends, etc., is misleading because it does not identify the native proteins. For instance, it should be clarified that p-Y1472 levels in Fig. 4 means p-GluN2B levels phosphorylated at Tyr 1472 (line 76). It may be changed to p-Y1472 GluN2B or similar. The same applies to the results obtained with the other antibodies of phosphorylated proteins. Check also Table 1 for the correct name of the antibodies.

6 – Please add kDa in all the showed blots in the paper figures. Western blot analysis are performed with the minimal number of samples for statistical analysis (n=3). Although results were clear here, increasing n may be generally desirable. Also, in supplementary information of Western blot membranes, it is expected to see the whole membrane as developed for each antibody, including the side indication of the protein ladder.

 7- Data not shown is not acceptable (line 310). The authors may give briefly the numbers into the text or add them as supplementary information.

8- Discussion should be more concise.  The introductory part to each subject is too long and partially overlaps with the Introduction section. Please shorten the discussion to aspects directly related to the determinations performed. In addition, there is no need to list and present all results again, delete references to figures.

9- An increased period of time spent in the center of the Open field (OF) is not per se an indication of anxiety (line 517), but rather of hyperactivity or disinhibition. To confirm anxiety changes, the authors may show the freezing time or the time of crossing the first quadrant.  Alternatively, there are specific behavioral tests for anxiety behaviors.

Author Response

The authors submitted an interesting experimental study about the neuroprotective effects of an extract of the plant Moringa Oleifera (MO) in APP/PS1 mice.

The study includes a wide number of techniques that as a whole demonstrated the multiple mechanisms activated by MO. All determinations are well done and results are interesting. However the manuscript should be edited to improve clarity and concretion.

Answer: Thank you very much for appreciating and commanding our research, and we are very grateful for the constructive observations and suggestions that have definitely helped to scientifically improve our manuscript.

  1. In the abstract, C57 (line 28, and also all over the text and figures) is not an accepted abbreviation and the full strain name should be used instead. Otherwise, WT abbreviation is introduced in line 95.

Answer: Thank you very much for this observation. We have now changed all the “C57” to “WT” throughout the text and in both the figures and figure legends. Please see highlighted in yellow in the text and in the figures in our revised manuscript.

  1. In Introduction, relevance of amyloid peptide generation for AD is addressed (lines 54 ss). However, please comment that there is a consensus in sporadic AD that an age-related deficiency of amyloid clearance has also a role in amyloid beta accumulation. Therefore inhibiting its generation may only partially contribute to an improvement.

Answer: We are very grateful for the expertise of the reviewer and constructive suggestions. We have now added “Moreover, it is believed that in sporadic AD, an age-associated decrease in Aβ clearance is also critical for Aβ accumulation [13-16]. Therefore, inhibiting Aβ generation and enhancing its clearance might synergically contribute to the improvement of AD” in the revised version of our manuscript. Please see highlighted in yellow in the introduction section.

  1. The phosphorylation sites analyzed from the diverse proteins are not introduced (only Y1472, line 76). Please add briefly to show relevance although there were no previous results in TgAD animals. That can then be reduced in Discussion.

Answer: We greatly appreciate this suggestion from the reviewer. We have now added “Fyn is the main kinase that phosphorylates GluN2B at Tyr1472 [36-38] and its activity is regulated by its phosphorylation at Tyr416. Interestingly, STEP can dephosphorylate (inactivate) Fyn at Tyr416 [39], thus impairing Tyr1472 GluN2B phosphorylation. On the other hand, in AD Aβ can induce an increase in STEP via impairing the ubiquitin-proteosome system as well as via activating the α7 nicotinic acetylcholine receptors (?7nAChRs) [40-42] which leads to intracellular calcium influx activating calcineurin (protein phosphatase 2B (PP2B)). The active PP2B can subsequently dephosphorylate (inactivate) the inhibitor of protein phosphatase 1 (PP1) DARPP-32 at Thr34, thereby leading to the dephosphorylation (activation) of PP1 at Thr320. Interestingly, PP1 can undergo auto-dephosphorylation and also can trans-dephosphorylate other PP1 molecules [43, 44]. Thus, upon removal of the inhibitory effect of DARPP-32, the active PP1 can then dephosphorylate (activate) STEP at Ser221 [32].” in the introduction section and have reduced it in the discussion part in our revised manuscript.

  1. There are some points to improve in the experimental design that the authors should consider in future work: (i) The use of both sexes is increasingly required in AD experimental research; (ii) It is recommended to add a WT treated group, please Indicate the known effects of MO treatment in the WT animals if available; (iii) Control WT and APP/PS1 must receive the vehicle (methanol) used in the MO-treated APP/PS1 animals. Please comment these deficiencies in limitations of the study.

Answer: We greatly appreciate the expertise of the referee in this regard. We take note and will consider these constructive suggestions in our future works. We have not found any available effects of MO alone on WT animals because it was always associated with other treatments. However, we made clear that MO is safe at high doses “and is safe at higher doses in both rats and mice as its LD50 was more than 6400 mg/kg in mice” in the introduction section, and we will consider a separate treatment group of MO in WT animals in our future works. We have also stated all these deficiencies in the last part of the discussion section as limitations “It should be noted that our experiment presents some limitations which include: (i) the use of male animals only in the experiment; (ii) no MO-treated WT animal group to explore the effect of MO in these animals; and (iii) no vehicle (methanol) control treatment in both the WT and APP/PS1 animals. These deficiencies will be considered in our future studies for better reliability of data, more originality, and standardization of experimental research”. Please see highlighted in yellow in the corresponding sections in our revised manuscript.

  1. In Results, the name of phosphorylated proteins in graphs titles, legends, etc., is misleading because it does not identify the native proteins. For instance, it should be clarified that p-Y1472 levels in Fig. 4 means p-GluN2B levels phosphorylated at Tyr 1472 (line 76). It may be changed to p-Y1472 GluN2B or similar. The same applies to the results obtained with the other antibodies of phosphorylated proteins. Check also Table 1 for the correct name of the antibodies.

Answer: Thank you very much for these observations. We have now replaced “p-Y1472” with “p-Y1472 GluN2B”; “p-Y416” with “p-Y416 Fyn”; “np-S221” with “np-S221 STEP”; “p-T34” with “p-T34 DARRP-32”; and “p-T320” with “p-T320 PP1” and explain their meaning in the result section and throughout the text. Where appropriate we have also added “p-Y1472=p-Y1472 GluN2B”; “p-Y416=p-Y416 Fyn”; “np-S221=np-S221 STEP”; “p-T34=p-T34 DARRP-32”; and “p-T320=p-T320 PP1” in the figure legend.

  1. Please add kDa in all the showed blots in the paper figures. Western blot analysis are performed with the minimal number of samples for statistical analysis (n=3). Although results were clear here, increasing n may be generally desirable. Also, in supplementary information of Western blot membranes, it is expected to see the whole membrane as developed for each antibody, including the side indication of the protein ladder.

Answer: Thank you for these observations. We have now added the kDa for the blotted proteins. Please see the new figures in the revised manuscript. We will consider increasing the “n” in our future works for better statistical reliability. We have also made sure that all the proteins are seen and that the side indications of the protein ladder are clear in the Western blot membranes in the supplementary information. However, some of the side indications of the protein ladder are not very clear due to the contrast adjustment. Please see the corresponding sections in our revised manuscript.

  1. Data not shown is not acceptable (line 310). The authors may give briefly the numbers into the text or add them as supplementary information.

Answer: Thank you for this suggestion. We have now added the cued memory test results in the supporting information. Please see “Supplementary Figure 1” under “Supporting information 8” in the revised supporting information.

  1. Discussion should be more concise.  The introductory part to each subject is too long and partially overlaps with the Introduction section. Please shorten the discussion to aspects directly related to the determinations performed. In addition, there is no need to list and present all results again, delete references to figures.

Answer: We are very grateful for these constructive suggestions. We have now shortened the introduction parts of all the sections in the discussion part and have only partially discussed some of the results. We have also deleted the references to figures. Please see the discussion part of the revised version of our manuscript.

  1. An increased period of time spent in the center of the Open field (OF) is not per se an indication of anxiety (line 517), but rather of hyperactivity or disinhibition. To confirm anxiety changes, the authors may show the freezing time or the time of crossing the first quadrant.  Alternatively, there are specific behavioral tests for anxiety behaviors.

Answer: We are very grateful for this observation from the referee. To avoid misunderstanding and over-interpretation of our results, we have now replaced the word “anxiety” with “anxiety-like behavior” in the revised version of our manuscript. Please see highlighted in yellow the corresponding section in the revised manuscript.

Round 2

Reviewer 1 Report

I think, the text was grossly revised as suggested. Because of a large number of molecular players, the DISCUSSION is still difficult to understand. However, the beautiful results would cover this drawback well.